# Mini-Review: Tregs as a Tool for Therapy—Obvious and Non-Obvious Challenges and Solutions

**DOI:** 10.3390/cells13201680

**Published:** 2024-10-11

**Authors:** Elena I. Morgun, Irina A. Govorova, Maria B. Chernysheva, Maria A. Machinskaya, Ekaterina A. Vorotelyak

**Affiliations:** Laboratory of Cell Biology, Koltzov Institute of Developmental Biology of Russian Academy of Sciences, 26 Vavilov Street, Moscow 119334, Russia; ischenko.i.a@gmail.com (I.A.G.); m.b.cher@gmail.com (M.B.C.); maria.machinsky@gmail.com (M.A.M.)

**Keywords:** Tregs, CAR-Tregs, FOXP3, immunotherapy

## Abstract

Tregs have the potential to be utilized as a novel therapeutic agent for the treatment of various chronic diseases, including diabetes, Alzheimer’s disease, asthma, and rheumatoid arthritis. One of the challenges associated with developing a therapeutic product based on Tregs is the non-selectivity of polyclonal cells. A potential solution to this issue is a generation of antigen-specific CAR-Tregs. Other challenges associated with developing a therapeutic product based on Tregs include the phenotypic instability of these cells in an inflammatory microenvironment, discrepancies between engineered Treg-like cells and natural Tregs, and the expression of dysfunctional isoforms of Treg marker genes. This review presents a summary of proposed strategies for addressing these challenges.

## 1. Introduction

Tissue-resident immune cells are vital for maintaining homeostasis in every tissue. Recently, the development of high-tech biomedical products based on cultured cells has led to novel approaches such as using immune cells. T regulatory cells (Tregs) have an impressive potential for immunotherapy applications. Tregs are a subpopulation of CD4+ T cells, and can modulate the immune system, maintaining tolerance to their own antigens and preventing autoimmune disease. The direct immunosuppressive mechanisms of Tregs can involve the interaction between CTLA-4 expressed on Tregs, and CD80 and CD86 on antigen-presenting cells. Indirect mechanisms involve the secretion of anti-inflammatory cytokines. Moreover, Tregs produce IL-10, TGF-β, and IL-35, as well as granzyme and perforin. These enzymes induce apoptosis in target cells, for instance, inflammatory cells [1,2].

The development of Tregs is regulated at the transcriptional level by the Forkhead box P3 (FOXP3) transcription factor, which is a member of the forkhead/winged-helix transcription regulator family. Tregs are absent in Scurfy mice with frameshift mutation in the *Foxp3*. This causes activation of effector T cells and, consequently, increased cytokine production [3]. Mouse *Foxp3* gene encodes a single protein product, while human *FOXP3* gene encodes two major isoforms through alternative splicing. The longer isoform (*FOXP3FL*) contains all coding exons, while the shorter isoform (*FOXP3ΔE2*) is devoid of the amino acids encoded by exon 2. These two isoforms are naturally expressed in humans. However, in inflammatory pathologies, the expression of the isoforms is dysregulated, either in favor of *FOXP3 FL* or in favor of *FOXP3ΔE2* [4] (Figure 1).

The potential therapeutic feasibility of using Tregs as a tool for therapy arises from the fact that Tregs are an essential component of the microenvironment in tissues. To illustrate this point further, Tregs activated as a result of the interaction of amphiregullin and EGFR on their surface play a role in maintaining skin homeostasis by suppressing inflammatory cells [5]. It has been shown that Tregs suppress the release of IL-17A by neutrophils, which causes the release of CXC chemokine ligand 5 (CXCL5) by damaged keratinocytes, which leads to the return of keratinocytes to the stem state and, as a result, delays reepithelization [6]. Additionally, Tregs exert a wound-healing effect through their influence on macrophages. It has been demonstrated that Foxp3-deficient transgenic *Foxp3^cre^Egfr^fl/fl^* mice exhibit an elevated percentage of proinflammatory macrophages during the initial stages of wound healing in comparison to wild-type animals [7]. As stated by Nayer and co-authors, Tregs have a beneficial effect on damaged skin tissues, as well as bones and muscles, through the modulation of monocytes/macrophages (Mo/MΦ). This process involves the conversion of these cells from a pro-inflammatory to an anti-inflammatory phenotype, which is mediated by the release of IL-10 [8]. There is evidence that a comparable mechanism underlying the ‘regenerative’ action of Tregs in affected tissues may also occur in central nervous system pathologies. A neuromyelitis optica spectrum disorder (NMOSD) model demonstrated that depletion of Tregs resulted in a reduction of IL-10 expression. Additionally, a decrease in the proportion of macrophages that expressed alternative activation markers, namely CD206 and Arg-1, was observed. Concurrently, the proportion of macrophages expressing classical activation markers (iNOS and CD16) was found to decline with the adoptive transfer of Tregs. Consequently, these findings suggest that Tregs influence macrophages/microglia towards an anti-inflammatory phenotype [9]. Furthermore, the regeneration of nervous tissue under the influence of Tregs can occur not only indirectly through macrophages, but also directly. In vivo experiments on Treg-depleted mice on a model of multiple sclerosis demonstrated that Treg cells play a pivotal role in remyelination of nerve fiber and oligodendrocyte differentiation through the production of CCN3, a multifunctional factor involved in regeneration [10,11].

Immunotherapy with Tregs represents a promising avenue for the treatment of a wide range of socially significant pathologies that significantly impair the quality of life and have limited treatment options [12,13]. It has been suggested that Tregs could be a potential therapeutic agent for the treatment of skin conditions such as vitiligo and alopecia, as well as psoriasis [2]. Moreover, the modulation of the cellular immune response appears to be an urgent area of research with the potential to yield new approaches in the treatment of central nervous system diseases, including Alzheimer’s disease [14,15,16]. Furthermore, immunotherapy can be utilized for the treatment of type 1 diabetes mellitus, bronchial asthma, and others [17,18,19,20].

At the same time, there are several challenges associated with the use of Tregs as a therapeutic tool. The most obvious issue is the non-selectivity of polyclonal Tregs. This challenge is currently being overcome through the generation of genetically engineered Tregs that are specific to a particular epitope of target tissues for homing and tissue-selective action. However, a multitude of additional, less frequently addressed obstacles also arise. In particular, when working with autologous Tregs, one significant challenge is the potential instability of their phenotype. This is due to the inflammatory microenvironment and mutations in the *FOXP3* gene. An alternative approach is the generation of genetically engineered Treg-like cells with permanent *FOXP3* expression. However, this approach presents a significant challenge for achieving a phenotype that closely resembles natural Tregs. This issue is purported to be addressed by additional genetic and epigenetic modifications of Treg-like cells. We discuss these issues in more detail below.

## 2. Non-Selectivity of Polyclonal Tregs

The use of polyclonal Tregs as a therapeutic tool has been the subject of active investigation in recent years. A number of both experimental and preclinical studies have been published [21,22,23]. Furthermore, several studies have reached the stage of clinical trial [24,25,26] (Table 1).

Tregs transplantation demonstrated an improvement in cognitive functions and a reduction in amyloid plaque accumulation in the triple-transgenic mice 3xTg-AD model [27] which exhibits excessive amyloid plaque and tau protein accumulation. It was demonstrated that pharmacological inhibition of Tregs activity was accompanied by the clearance of amyloid-β plaques, mitigation of the neuroinflammatory reaction, and reversal of cognitive decline in the 5xFAD transgenic mouse model [28]. Furthermore, the possibility of a binary therapy by combining Tregs with other cell types has also been demonstrated. Chen and co-authors demonstrated that mesenchymal stem cells (MSC) derived from umbilical cord can play a pivotal role in the treatment of neurological disorders following brain injury which is achieved by maintaining the balance of Th17/Treg cells [21]. Additionally, in vitro study on rat splenocytes and human peripheral blood mononuclear cells (PBMCs) demonstrated that combined therapy with Tregs and bone marrow-derived MSC resulted in the reduction of IFN-γ and TNF-α production. In other words, combined Treg cell and MSC therapy has an advantage over monotherapy in improving the immune response [22]. However, no significant benefits on blood–brain barrier integrity and immune response of binary therapy over monotherapy were identified in vivo after traumatic brain injury (TBI) in rats [23].

Potential therapeutic applications of Tregs in the treatment of type 1 diabetes mellitus have also been studied. The investigation of polyclonal Tregs in the treatment of type 1 diabetes mellitus has now reached the clinical trials stage. A phase 1 safety assessment study of adoptive immunotherapy with Tregs in type 1 diabetes demonstrated that ex vivo-extended autologous polyclonal CD4(+)CD127(lo/−)CD25(+) Treg cells retained their original properties and viability. Moreover, their infusion did not cause any adverse effects in patients. However, no notable metabolic improvements were reported. C-peptide levels remained elevated for a period of two years or more following the transfer, while HbA1c levels remained stable in all but one subject [24]. A trial of expanded autologous Tregs in children with type 1 diabetes (dose of Tregs = 30 × 10⁶ Tregs/kg) for a period of two months following diagnosis resulted in some improvement in the functioning of beta cells after infusion. This was evidenced by the prolongation of remission of the disease. Most patients exhibited an increase in C-peptide levels in response to therapy. The results demonstrated that 8/12 patients exhibited this response following the initial dose and 4/6 patients following the second dose. Additionally, the administration of Tregs resulted in a reduction in the requirement for exogenous insulin (8/12 patients compared to 2/10 patients in the untreated group), with 2 patients remaining insulin independent for a period of one year. However, despite these promising initial outcomes, the disease ultimately progressed, resulting in insulin dependence for all patients within two years [26].

There are certain limitations to the efficacy of polyclonal Treg cell therapy. Firstly, a substantial number of cells must be administered to achieve the desired outcome. Consequently, Tregs limit the functions of effector cells, and prolonged disruption to this process may result in the persistence of the pathogen. This issue can be addressed through the administration of Tregs following the removal of the pathogen [29]. Furthermore, nonspecific immunosuppression may potentially contribute to cancer development [30]. It is possible to resolve this challenge by employing a smaller number of Tregs that have the capacity to migrate specifically towards the site of inflammation and exert an immunosuppressive effect, as opposed to polyclonal cells [31].

The initial Treg-based approaches were tested in transplantation models, and a number of studies have employed stimulation of polyclonal Tregs using donor antigen-presenting cells (APC) in order to increase the number of Tregs specific to allogeneic antigens. [31,32,33]. In a model of human skin xenotransplantation in NOD/scid/*IL-2R*γ^−/−^ and BALB/c*Rag2*^−/−^γ*c*^−/−^ mice, administration of Tregs specific for allogeneic antigen significantly decreased clinically relevant markers of skin tissue damage compared to polyclonal Tregs. Moreover, it was shown that the tissue morphology exhibited restoration comparable to that observed in healthy skin [31]. Antigen-specific Tregs demonstrated migration towards the site of inflammation and caused immunosuppression via interaction between T cell receptor (TCR) and main histocompatibility complex class II (MHC II) molecules [34]. It is, however, possible to create genetically engineered Tregs carrying chimeric antigen receptors (CAR) in an MHC-independent manner. CAR comprises two distinct domains: an antigen-binding domain and a signaling domain. These domains are essential for genetically modified Tregs to recognize and attach to a target cell, and to become activated [35,36]. Consequently, antigen-specific CAR-Tregs migrate to their target antigen and attach to the cells of the damaged tissue, thereby suppressing the anti-inflammatory milieu [2].

A number of studies have been focused on the creation of CAR-Tregs for the treatment of various pathologies, including type 1 diabetes, asthma, hemophilia, graft-versus-host reactions and others, using animal models [17,37,38,39]. The selection of an appropriate target is of vital importance when designing a CAR. For example, insulin-specific CAR-Tregs have been demonstrated to be capable of migrating to the pancreas, exerting an immunomodulatory effect at the site of tissue destruction in type 1 diabetes mellitus [37,40]. It has been demonstrated that CAR-Tregs specific to the peptide 10–23 of the insulin B chain had a more powerful therapeutic effect than polyclonal cells in the treatment of type 1 diabetes mellitus in NOD mice [37].

The efficacy of CAR-Tregs therapy for the treatment of inflammatory skin pathologies, including vitiligo and alopecia, is discussed. Mukhatayev and co-authors demonstrated the efficacy of CAR-Tregs targeting ganglioside D3, which is overexpressed by epidermal cells, including melanocytes, and suggested a potential use of CAR-Tregs in the treatment of focal alopecia [2]. It is proposed that melanocyte epitopes, for example, tyrosinase (TYR), tyrosinase-related protein-1 (TRP-1), TRP-2, and glucoprotein 100 may be the target for CAR-Tregs in the treatment of focal alopecia. In addition, several other epitopes have been identified as potential targets for CAR-Tregs: melan-A and melanocortin-1-receptor (MC1R), as well as keratinocyte epitopes such as trichogyalin and keratin [41]. It is also suggested that CAR-Tregs may be used in the treatment of other inflammatory skin pathologies including scleroderma, pemphigus, pemphigoid, and psoriasis [2].

A model of asthma was established using carcinoembryonic antigen (CEA) transgenic mice. CEA is a glycoprotein present on the surface of the adenoepithelium of the lungs and gastrointestinal tract. In transgenic CEA mice with asthma, CEA-specific CAR-Tregs accumulated and were activated in the inflamed lungs. It was demonstrated that CEA-specific CAR-Tregs were more effective in controlling key symptoms of allergic inflammation in much smaller amounts compared to unmodified Tregs [17,42]. The transfer of 5 × 10⁶ unmodified, in vitro-multiplied CD4+CD25+ Tregs successfully suppressed respiratory tract inflammation in a model of allergic asthma [42]. Meanwhile, the introduction of 1 × 10^6^ CAR-Tregs significantly reduced experimental asthma. It can be further proposed that CAR-Tregs technology may allow for the administration of a reduced number of cells in comparison to polyclonal Treg therapy [17].

Consequently, the use of specific CAR-Tregs overcomes a challenge associated with polyclononal Tregs, namely their non-specificity. This enhances the effectiveness of Treg therapy, allows for a much smaller number of cells, and is also safer for the patient.

## 3. Instability of the Phenotype of Autologous Tregs

Another challenge associated with the use of either polyclonal or specific Tregs as a therapeutic tool is the maintenance of their phenotypic stability. It is known that conventional Tregs have the ability to convert into pro-inflammatory Th-like Tregs and Th17-like Tregs in autoimmune conditions [43]. It was demonstrated that brain Tregs exhibit an unstable phenotype and are unable to control reactive gliosis in response to repeated stimulation by viral antigen peptides. The number of CD4+ cells with FOXP3 expression in the brain was observed to decrease upon repeated infection with viral antigen peptides. When exposed to repeated viral antigen, brain Tregs expressed less amphiregulin (Areg) and suppression of tumorigenicity 2 (ST2) and had an ineffective immunosuppressive potential [44].

Despite the number of works reporting that the phenotype of transplanted Tregs is relatively stable, there are some concerns about Treg phenotype instability in the context of inflammatory diseases [24,45,46,47]. It was demonstrated in clinical studies that following the initial infusion of Tregs to children with type 1 diabetes mellitus, a notable shift occurred from the naive CD62L+CD45RA+ phenotype of Tregs to the CD62L+CD45RA− phenotype of central memory Tregs. The study discovered a slow increase in pro-inflammatory activity, which was only partially controlled by Tregs administration [25]. Furthermore, it has been shown that prolonged type 1 diabetes mellitus can result in alterations in Treg properties, which may be attributed to the inflammatory milieu. Under proinflammatory conditions, these cells demonstrate diminished FoxP3 and CD62L expression which can be reversed by anti-inflammatory drugs [48,49,50].

It is thought that an imbalance between the expression of splicing isoforms of the *FOXP3* gene may be a contributing factor to the instability of the Treg phenotype observed in inflammatory pathologies. In patients with vasculitis, Hashimoto’s thyroiditis, giant cell arteritis and celiac disease, the expression of the *FOXP3ΔE2* isoform increases, becoming the dominant isoform, which suggests a possible correlation of this isoform with autoimmune diseases [51,52,53,54]. Conversely, in patients with rheumatoid arthritis, the ratio of *FOXP3ΔE2* mRNA to *FOXP3FL* is reduced compared to healthy control groups [55]. Similarly, the expression of *FOXP3FL* mRNA, but not *FOXP3ΔE2*, is significantly increased in PBMCs of patients with coronary heart disease [56]. In murine Tregs, it was demonstrated that the expression of *Foxp3Δ2*, in lieu of *Foxp3FL*, results in the loss of crucial phenotypic markers, including CD25, FOXP3, and CTLA-4. Furthermore, the transfer of *Foxp3Δ2*-expressing Tregs into *Tcrb*-deficient mice has been shown to induce autoimmunity [4]. The lack of FOXP3 expression in these cells results in the loss of their suppressive function and, to some extent, the acquisition of the phenotypic and functional characteristics of effector T helper cells. This is accompanied by the production of IFN-γ and IL-17 [57]. Therefore, it is assumed that an imbalance in the expression of FOX3 isoforms is associated with autoimmune diseases.

The underlying causes of the observed alterations in the ratio of *FOXP3* isoforms in autoimmune disorders remain uncertain. In IPEX syndrome, a condition associated with type 1 diabetes mellitus, atopic dermatitis, and refractory diarrhea, dysfunctional Tregs are characterized by a mutation in the second exon of the *FOXP3* gene (305delT). This alteration results in the expression of *FOXP3Δ2* instead of *FOXP3FL* [4]. This corroborates the hypothesis that the ratio of FOXP3 isoforms is subject to change in the context of genetic abnormalities. Conversely, it was demonstrated that exposure to IFN-γ and butyrate can result in a shift in the ratio of *FOXP3ΔE2:FOXP3FL* expression in PBMCs. This suggests that the balance of the two *FOXP3* isoforms may be influenced by both genetic and microenvironment factors [54]. Nevertheless, the transdifferentiation of Tregs into effector T cells may potentially compromise the intended therapeutic effect.

This issue may be mitigated through the generation of genetically stable cells, designated as Treg-like cells. This can be achieved by the transduction of CD4+ T conv cells with *FOXP3* gene, for instance, using recombinant adeno-associated serotype 6 virus (rAAV6) or lentiviral vectors [58,59]. Tenspolde and co-authors demonstrated the generation of functionally active insulin-specific CAR-Tregs using lentiviral CD4+ transduction of mouse effector T cells. This approach involved the use of a system of chimeric antigen receptor vectors, which contained both an insulin-specific CAR and the Foxp3 gene. Stable expression of Foxp3 in these cells resulted in transformation of effector T cells into Treg-like cells. Insulin-specific Treg-like cells exhibited phenotypic and functional similarities to native Tregs. Moreover, it was established that the obtained cells demonstrated a long life expectancy of up to four months in a model of type I diabetes in NOD/Ltj mice [40]. Consequently, it can be assumed that ectopic expression of *FOXP3* in CD4+ T conventional cells not only contributes to the formation of a Treg-like phenotype in these cells, but is also compatible with the production of CAR-Tregs.

## 4. Approximation of Genetically Engineered Treg-like Cells to Natural Tregs

Multiple studies have demonstrated the feasibility of generating Treg-like cells by ectopically expressing *FOXP3*. Nevertheless, the results pertaining to the phenotypic and immunosuppressive properties of these cells remain inconclusive and contradictory. Goodwin and co-authors have shown that engineered Treg-like cells demonstrated equivalent expression levels for the basic Treg markers, including TIGIT, HELIOS, PD-1, CTLA-4, and CD25, as observed in native Tregs [58]. Furthermore, Sato and co-authors reported that Treg-like cells exhibited elevated levels of CTLA-4, ICOS, GITR, IL-1R1, and IL-6R, as well as Treg-related gene markers *FOXP3*, *LGMN*, *EOS*, *IL-12A*, and *EBI3*, indicating an enhanced Treg phenotype [59]. Tenspolde and co-authors showed that insulin-specific CAR-Treg-like cells in vitro expressed several important Treg markers, including FOXP3, CD25, GITR, and CTLA-4. Furthermore, the cells also expressed the Treg activation markers CD62L and CD69 [40]. However, despite these data, discrepancies in the expression profiles of Treg marker genes within Treg-like cells were found. For instance, Sato and co-authors report no difference in the number of crucial Treg markers, such as HELIOS, PD1, and LAG3, between Treg-like cells and T conventional cells transduced with the control plasmid [59]. In the study of Tenspolde and colleagues, Treg-like cells were observed to lack the expression of HELIOS [40].

There is compelling evidence that engineered FOXP3+ cells can acquire Treg-like properties. This has been demonstrated by their capacity to suppress CD4 T+ killer and helper cells, as well as their cytokine profile, which is characterized by reduced proinflammatory cytokine production (e.g., IL-2, IL-4, IL-17A, and IFN-γ) and preserved the production of anti-inflammatory cytokines (such as IL-10 and IL-22). These characteristics are comparable to those of in vitro-activated Tregs [40,58,59,60]. Furthermore, the polyclonal repertoire of TCR in vitro was also preserved [59]. However, Gallego-Valle and co-authors report that, despite the expression of Treg markers CD25, FOXP3, CTLA-4, and CD39, engineered *FOXP3+* cells were found to be unable to induce a robust immunosuppression function, as well as express pro-inflammatory cytokines IL-2 and IFN-γ. Despite the administration of TGF-β1, interleukin-2 (IL-2) and CD3/CD28, which are known to stimulate Tregs, this effect was not observed [61].

It can be concluded that ectopic expression of *FOXP3* in CD4+ T conventional cells only partially contributes to the replication of the Treg phenotype. This presents a challenge for the generation of CAR-T cells with Treg-like properties, namely, the modifications required for the engineered FOXP3+ T cells to be as close to the Treg phenotype as possible.

It is hypothesized that stable imprinting of the Treg phenotype is mediated by demethylation of the conserved non-coding sequence 2 (CNS2) region of the *FOXP3* gene, which is characteristic of the natural Tregs. This epigenetic regulation supports the expression of FOXP3 protein [62,63]. Chen and co-authors report that the combination of three chromatin-modifying chemical compounds (3C) prevented histone deacetylation and methylation, as well as promoted DNA demethylation in the CNS2 region of the *Foxp3* locus, which resulted in a stable Treg phenotype. It was demonstrated that the obtained Tregs prevented the induction of autoimmune encephalitis and enhanced the viability of skin allografts in murine models of autoimmunity and transplantation [64]. Concurrently, it was demonstrated that demethylation of CNS2 did not result in the stabilization of FOXP3 expression. However, the introduction of cells modified with dCas9 fused to the CD of p300, a histone acetyltransferase (dCas9p300CD), with a guide RNA targeted to the *Foxp3* gene locus, resulted in a sustained expression of Treg markers CD25 and CTLA-4, as well as an augmented suppressive capacity in vitro [65].

Therefore, epigenome editing represents a promising approach to approximating the phenotype of engineered Treg-like cells to that of native Tregs. However, further studies are required to fully assess the efficacy of this method.

It has been demonstrated that there are alternative methods for enhancing the Treg-like properties of CD4 T lymphocytes. It was observed that the transduction of *FOXP3^−^/^−^* T cells with a lentiviral vector expressing two cDNA constructs based on *FOXP3* isoforms (*FOXP3FL* and *FOXP3Δ2*) led to a more durable acquisition of regulatory T cell-like properties compared to the transduction of individual isoforms separately. This was evidenced by a higher expression of FOXP3 protein by cells transduced with two isoforms, and by an increase in suppressive function and a decrease in the expression of inflammatory cytokines IL-17A and IL-22 compared with cells transduced with *FOXP3FL* and *FOXP3Δ2* individually. In *FOXP3FL+FOXP3Δ2* cells, the suppressive activity was comparable to that observed in wild-type Tregs [66]. Thus, the development of genetically engineered Treg-like cells for CAR-Treg therapy would be more efficacious if conventional T cells were transduced with a vector containing cDNA based on two splice forms of the *FOXP3* gene, rather than the *FOXP3* gene itself.

In addition to the transduction of T cells using *FOXP3* isoforms, it is possible to enhance the Treg-like properties of transduced T cells by introducing isoforms of other Treg crucial genes, such as *HELIOS*, into them. At the present time, the role of HELIOS in supporting the Treg phenotype is under discussion. However, there is a prevailing view that HELIOS is associated with Treg stability. It has been demonstrated that Tregs derived from individuals with type 1 diabetes mellitus display reduced HELIOS expression during in vitro expansion, potentially indicating a loss in phenotype stability [67]. Conversely, there is evidence that *HELIOS* does not directly contribute to maintaining stability. This is evidenced by the observation that cells lacking this gene exhibited no difference in suppressive activity and stability under inflammatory conditions [68]. Concurrently, HELIOS is demonstrated to bind to the *FOXP3* promoter, enhancing *FOXP3* transactivation, and to disable the IL-2 gene promoter, contributing to the development and stability of Tregs [69].

It is known that *HELIOS* has two mRNA isoforms: the full-length mRNA isoform (*Hel-FL*) and the mRNA isoform that lacks part of the third exon (*Hel-Δ3B*) [70]. Seng and co-authors conducted a retroviral transduction of CD4+ and CD8+ T cells not only with *FOXP3*, but also with *Hel-FL* or *Hel-Δ3B*. In vitro, cells obtained by the co-transduction of *FOXP3* and *Hel-FL* suppressed T conventional cells; the effect was mirrored in vivo in the graft-versus-host reaction induced by injection of the human blood mononuclear cells into mice. The transduction of CD4+ and CD8+ T cells with *FOXP3* and *Hel-Δ3B* did not yield comparable results. RNA sequencing demonstrated that the co-expression of *Hel-FL* or *Hel-Δ3B* with *FOXP3* led to an increased expression of genes associated with Treg signatures in the resulting cells when compared with T cells transduced solely with *FOXP3*. Collectively, these findings demonstrate that functional human CD4+ Treg-like cells can be derived from human T conventional cells through the co-expression of *FOXP3* and *Hel-FL*. Additionally, it was shown that the expression profile closest to the Treg signature was observed precisely during the ectopic expression of *FOXP3* and *Hel-Δ3B* in T conventional cells [71]. However, the possibility of combined transduction of T cells with both *HELIOS* isoforms, along with *FOXP3*, has not been investigated.

Another crucial regulator of Treg suppressive activity is CTLA4. For instance, CTLA4 is essential for Tregs to bind indirectly to CD80 and CD86 on antigen-presenting cells, which transmits an inhibitory signal [72]. *CTLA4* generates four splice variants: a full-length transcript (*CTLA4FL*) and three shorter transcripts lacking exons 2 and/or 3. The shorter mRNAs—*CTLA4* (*liCTLA-4*), 1/4 *CTLA4*—encode ‘ligand-independent’ proteins, and *sCTLA4* encodes the soluble form of CTLA4 [73]. The work of Kaartinen and others has demonstrated that *CTLA4FL* and *sCTLA4* isoforms are expressed in Tregs. It was shown that there was an increase in the proportion of *CTLA4FL* as Treg cells were cultured, which correlated with an increase in their suppressive activity [74]. However, Gerold and co-authors demonstrated that there was an observable reduction in the activity of Treg cells in mice with the expression of only one of the two splice forms, *sCTLA4*, suppressed. The absence of *sCTLA4* led to an acceleration of the development of autoimmune diabetes, thus confirming the role of this splicing variant in the pathogenesis of such diseases in humans. Consequently, the regulatory role of the *sCTLA4* isoform in Treg activity is of significant importance [75].

It is thought that the study of alternative splicing of genes crucial for Tregs in CD4+ T conventional cells is critical for the generation of Treg-like cells. Modulating alternative splicing of exons into the mRNA of the corresponding genes using splicing-switching oligonucleotides or gene editing using CRISPR/CAS9 may be a promising strategy for obtaining functional Treg-like cells [76].

Concurrently, it is reasonable to posit that specific domains within CTLA4 may enhance the quality of Treg-like cells. The CTLA4 protein is known to have three domains: the immunoglobulin V domain, the transmembrane domain, and the cytoplasmic region [77]. It is demonstrated that constitutive expression of only the extracellular domain of CTLA4, which determines the immunosuppressive properties of Tregs, is sufficient for the acquisition of Treg-like properties by CD4+ T cells, notably in the absence of FOXP3 [77,78]. At the same time, treatment with the recombinant cytoplasmic domain of CTLA-4 conjugated to cell-penetrating peptide (dNP2-ctCTLA-4) was shown to be effective in decreasing the progression of psoriatic skin inflammation and increasing the population of FOXP3+ CD4 T cells in a murine model of psoriatic inflammation. In vivo experiments revealed that this peptide can induce the expression of *FOXP3* in both CD4+ T cells and CD8+ T cells. This was achieved by inhibiting p-ERK and p-SMAD2/3 linker region in vitro. In addition, Treg cells induced by the dNP2-ctCTLA-4 peptide showed increased expression of functional markers such as CD25 and CD39, which suggests the induction of potent suppressive Treg cells. Through its binding to the PKC-n protein, the dNP2-ctCTLA-4 peptide enhances the TGF-β-Smad2/3 signaling pathway, leading to an increase in Foxp3 expression in CD4 T cells. It was demonstrated that dNP2-ctCTLA-4 is capable of effectively suppressing inflammation and mitigating pathogenic effects. This is achieved by inhibiting the production of IL-17A by Th17, Tc17, and γδ T cells [79]. At the same time, no attempt has been made to integrate the intracellular domain of CTLA-4 into cells. Perhaps, even though the outer domain of CTLA-4 exerts immunosuppressive activity, the introduction of the intracellular domain of CTLA-4 might bring genetically engineered Treg-like cells closer to the natural Treg pathway.

Therefore, in addition to the *FOXP3* gene, other Treg marker genes, their splicing variants and sequences coding for individual domains could be introduced in CD4+ T cells to generate Treg-like cells.

A summary of the challenges and potential solutions for the use of Tregs as a therapeutic tool is presented in Table 2.

## 5. Conclusions

The generation of therapeutic products based on Treg cells is a promising direction in biomedicine, but it is not yet widespread. In part, this is because not all the challenges of using these cells as a therapeutic tool are apparent today. The non-specificity in the use of polyclonal Tregs, which is resolved by the transduction of a specific CAR, is not the only problem in the creation of a therapeutic product based on these cells. An important dilemma is which cells are better suited to act as the basis for CAR Treg—native Tregs, which may change their phenotype in the inflammatory microenvironment (or due to FOXP3 dysfunction), or artificial Treg-like cells, in which ectopic expression of FOXP3 does not ensure full representation of Treg properties by the obtained cells. Approximation of the phenotype of Treg-like cells to native Tregs can be achieved by epigenetic modifications of FOXP3, as well as by cell transduction not only with Treg marker genes, but also with various combinations of their splice isoforms and individual domains. We suggest that the decision as to which CAR-Tregs to choose—those based on Tregs or those based on Treg-like cells—as well as the precise way in which Treg-like cells are generated, should be made considering the condition of the Tregs and their marker genes in any given patient.

## Figures and Tables

**Figure 1 cells-13-01680-f001:**
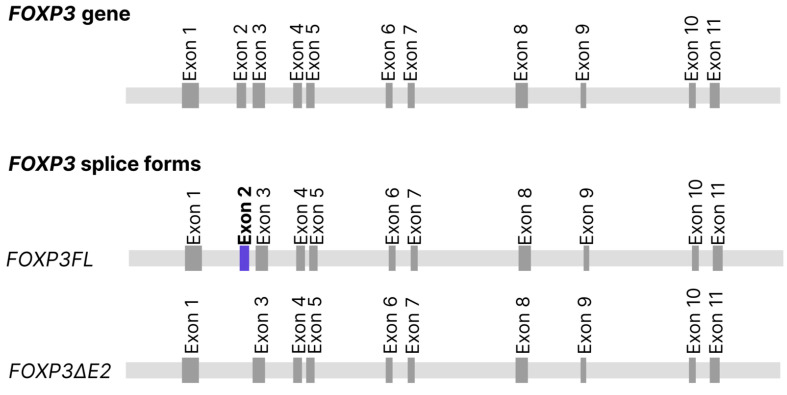
Schematic representation of the FOXP3 gene and its splice forms.

**Table 1 cells-13-01680-t001:** Clinical trials based on Tregs for the treatment of diabetes mellitus.

Author, Year	Number of Patients	Age	Number of Treg	The Scheme of Administration	C-Peptide	HbA1c	Response to the Therapy
[24]	26	30.3 ± 8.7 years	0.05 × 10^8^ − 26 × 10^8^	unspecified	increased	remained stable in all but one subject	not significant
[26]	12	5–18 years	10 × 106 of Tregs/kg	single infusion	increased	decreased	favorable
			20 × 106 of Tregs/kg	single infusion			
			30 × 106 of Tregs/kg	two infusions			
[25]	12	5–18 years	10 × 106 of Tregs/kg	single infusion	increased	decreased	low
			20 × 106 of Tregs/kg	single infusion			
			30 × 106 of Tregs/kg	two infusions			

**Table 2 cells-13-01680-t002:** The challenges and potential solutions for using of Tregs as a therapeutic tool.

Challenges	Solutions
The non-selectivity of polyclonal Tregs High doses of cells required for effective targeting (technical and safety issues)	Generation of CAR-T cells specific for antigens in target tissues, which can be effective at significantly lower doses
A phenotypic shift in the Tregs in an inflammatory milieu	Generation of Treg-like cells by transduction of CD4+ T cells with *FOXP3*
Incomplete correspondence between the engineered Treg-like cells and natural Tregs	Introduction of Treg marker genes and isoforms or individual domains into CD4+ T cells Epigenetic modifications of *FOXP3* Search for additional regulatory options
The expression of dysfunctional isoforms of Treg marker genes	Use of splicing-switching oligonucleotides or CRISPR/CAS9-mediated gene editing

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
