# Peer review of "Mini-Review: Tregs as a Tool for Therapy—Obvious and Non-Obvious Challenges and Solutions"

_cells, 2024, doi:10.3390/cells13201680_

Round 1
Reviewer 1 Report
Comments and Suggestions for Authors
In this review article authors discuss the utility of engineered T regulatory cells in various conditions and pathologies. Overall the article is well written following are some comments.
1. In the introduction authors should briefly describe the canonical utility of T regulatory cells in diverse pathophysiology. Also, apart of transcription factors it would be ideal to elaborate the mechanism of action of immunosuppresion i.e. via surface and soluble proteins and its utility in diverse pathophysiologies.
2. It would be ideal to have some figures for the introduction and for utility of T regulatory cells for cell therapy.
3. Authors should provide tabular version of key clincal trials based on T regulatory cells as describe in text.
4. Authors should discuss the proposed/speculated interventions to mitigate the adverse effect of T regulatory cell therapy
Comments on the Quality of English LanguageMinor corrections are needed overall well written.
Author Response
1. In the introduction authors should briefly describe the canonical utility of T regulatory cells in diverse pathophysiology. Also, apart of transcription factors it would be ideal to elaborate the mechanism of action of immunosuppresion i.e. via surface and soluble proteins and its utility in diverse pathophysiologies.
Thank you for pointing on this. In the introduction, we included additional information concerning Treg functions and mechanisms of immunosuppression including those via surface and soluble proteins. Meanwhile, exact mechanisms of Treg action are often illusive. It would be ideal to have some figures for the introduction and for utility of T regulatory cells for cell therapy.
2. It would be ideal to have some figures for the introduction and for utility of T regulatory cells for cell therapy.
We agree that it would be useful to include this in the review, however, due to limited time for editing, unfortunately, we were not able to create qualitative and informative figures and decided better not to do it.
3. Authors should provide tabular version of key clincal trials based on T regulatory cells as describe in text.
Thank you for the recommendation. We have added a table of key clinical trials presented in the review to the text on page 3.
4. Authors should discuss the proposed/speculated interventions to mitigate the adverse effect of T regulatory cell therapy
It is hypothesized that the principal adverse effects of polyclonal Tregs, derived from their suppressive functions, may include the persistence of the pathogen and the development of a cancerous tumour. We highlight that the initial issue is addressed through the utilisation of Tregs following preliminary infection treatment. The subsequent issue is resolved through the reduction of the administered Treg count during therapy. This necessitates the generation of antigen-specific Tregs, which can be administered in reduced quantities and will act locally. The requisite amendments have been incorporated into page 4, paragraph 3.
5. Comments on the Quality of English Language Minor corrections are needed overall well written.
We have also improved the quality of English in the text according to your advice

Reviewer 2 Report
Comments and Suggestions for Authors
It a very well written and comprehensive review adressing the major issues with the clinical application of Tregs in a number of autoimmune diseases. Th problem of Treg non-selectivity as well as of the phenotypic instability especially in the context of an inflammatory microenvironment are discussed in detail.
Minor comments
Improve the quality of the graphics (Tables)
Lines 112-113 : the term drug resistance should be better replaced by the term resistance to treatment since Tregs are not a drug at least with the tradidtional view
Author Response
Thank you very much for appreciating our work. We have improved the quality of the table
Lines 112-113 : the term drug resistance should be better replaced by the term resistance to treatment since Tregs are not a drug at least with the tradidtional view
We completely agree, the term "resistance to treatment" is indeed more appropriate. However, in the course of our edits, on the advice of the first reviewer, we made significant alterations to this paragraph, which resulted in the complete omission of this phrase.

Reviewer 3 Report
Comments and Suggestions for Authors
The authors' goal is to present the current status of Treg cell therapies and explore the advantages and disadvantages of each treatment option.
They first describe the current status of polyclonal Treg cell therapies, highlighting in particular the advantages and disadvantages in mouse models and in type 1 diabetes mellitus. They then present CAR Treg treatments for skin diseases and asthma. Once more, they discuss the drawbacks in conjunction with the advantageous clinical outcomes.
Next, they delve into another concern associated with Treg cell therapies. This is the loss of phenotype stability. They present the topic of Treg-like cell generation as a solution, following a concise presentation and critique of the clinical trials. They provide a detailed description of the methodology and results of the available studies. They summarize the challenges and potential solutions for Treg therapies in a clear and comprehensible manner.
The article is well-written, concise, and an understandable short review. The references are relevant.
They assist not only researchers but also clinicians in selecting optimal Treg cell therapies.
The article meets Cells' strict publication criteria in terms of content, style, and form.
Before accepting the article, correct the following typos:
L59: dueto
L123-124 reference form.
Author Response
Thank you very much for appreciating our work. We have made the edits suggested by you
